# Water–Land–Food Nexus for Sustainable Agricultural Development in Main Grain-Producing Areas of North China Plain

**DOI:** 10.3390/foods12040712

**Published:** 2023-02-06

**Authors:** Lijia Zhu, Yuping Bai, Lijin Zhang, Wanyi Si, Anni Wang, Chuyao Weng, Jiayao Shu

**Affiliations:** 1School of Land Science and Technology, China University of Geosciences, Beijing 100083, China; 2Institute of Geographic Sciences and Natural Resources Research, Chinese Academy of Sciences, Beijing 100101, China

**Keywords:** water–land–food nexus, coupling coordination, available water and land resources, food security, main grain producing areas of North China

## Abstract

Stable and sustainable food production is an important guarantee for national security and social stability. The uneven distribution of cultivated land and water resources will threaten national food security. In this study, we adopt the Gini coefficient and water–land matching coefficient for exploring the water–land nexus in the main grain-producing areas of North China Plain (NCP) from 2000 to 2020. The water–land–food nexus considering grain crop production structure is further explored from spatial and temporal multi-scales. The results show the following: (1) The Gini coefficient presents an increasing trend in the NCP, indicating an increasing imbalance in the water–land matching degree among inter-regions. (2) There are significant differences in the WL nexus and WLF nexus among regions, showing a spatial pattern of “worse in the north and better in the south”. (3) The cities which belonged to the low WL-low WLF and high WL-low WLF should be considered as key targets when formulating policies. (4) Adjusting the wheat–maize biannual system, optimizing the grain cultivation structure, promoting semi-dryland farming, and developing low water-consuming and high-yielding crop varieties are essential measures for these regions. The research results provide significant reference for the optimal management and sustainable agricultural development of agricultural land and water resources in NCP.

## 1. Introduction

Food provides a vital basis for a country’s long-term peace and stability. Water and cultivated land resources are the most important resources to support food production. Chinese Central Government Document No. 1 in 2021 highlighted that the key point to guarantee the national food security is to lay a solid substance basis using the strategy of “having grain in the ground” and “having grain in technology” in the new era. Because land and water resources are the two major rigid constraints influencing grain production [1], exploring how to effectively ensure the security of food supply under the constraints of water and land resources is an important realistic proposition that needs to be solved urgently. Moreover, it has been estimated that the shortage and the spatial misallocation of water and land resources directly affect the regional grain production capacity, and fundamentally restrict the development of agricultural modernization [2,3]. According to the report released by FAO, over the past decade, the state of the planet’s soil, land, and water resources has decreased dramatically, consequently increasing pressure to meet the food needs of approximately 10 billion people worldwide by 2050 [4]. Water, land, and food resources are the foundation for social stability, human survival, and economic development. Therefore, it is of great significance to study its water–land–food (WLF) nexus in order to promote the sustainable production of regional grain, sustainable utilization of land and water resources, as well as sustainable development of the economy [5].

China is a large agricultural producer, and agricultural production is associated with national food security. Nevertheless, China has 130 million hectares of cultivated land, which is only 7% of the world’s cultivated land resource and feeds 22% of the world’s population [6]. The normal annual total water resource of China is about 2840.5 billion m^3^, ranking sixth in the world [7]. However, the average per capita water resource in China is less than one-third of the world average [8]. Studies have revealed that the northern part of China has about 60% of the population [9], 64% of total land area, and 46% of cultivated land area, whereas water resources there occupy just 19% of the total water resources of China [10,11]. This means that the spatial imbalance between water resources and other social resources in China is an important, realistic proposition. At present, a severe water shortage has made agriculture in China face immense challenges [12]. Additionally, the spatial distribution of China’s water, land, and food resources is unbalanced [13]. As significant resource elements, water, land, and food are highly related, they are often regarded as a coupling system with mutual correlation, which is also called the WLF system for short. Moreover, it is of great practical significance and scientific value to identify the relationships among the WLF system.

The main grain-producing areas of North China are located in the north China Plain, which is one of the typical irrigation areas [14]. Its irrigation water consumption is high, and the irrigation water is almost obtained through groundwater exploitation. However, groundwater has been severely over-extracted, which has caused the largest underground water funnel area globally [15,16]. The study area is one of the biggest and most important commodity grain production bases in China. Therefore, the increasing intention of water shortage or the mismatch in spatial distribution between water and land resources can be regarded as rigid restraints of agricultural development [17,18], seriously affecting the food quality and security of China. As a result, many studies have been conducted on the two-dimensional relationship between water and land [19,20,21,22]. Using a nexus approach, resource allocation could be optimized more effectively. Most scholars take regional cultivated land or potential cultivated land resources and agricultural water resources as the research objects in order to calculate the water–land matching degree [1,23]. Based on the conception of blue water and green water, Geng et al. examined the abundance or deficiency of agricultural water and land resources [24]. Wu and Bao were the first to calculate the matching coefficients of water and land resources between provinces in China and other countries worldwide by developing a Lorenz Curve of agricultural water resources and cultivated land area [25]. Scholars usually adopt Gini coefficient method and the water–land resources utilization coefficient for identifying the matching pattern and evolution characteristics between two resources [26,27] Based on the binary relation between water and land resource, the multiple correlations with other resources including energy and carbon can be further explored [5,28]. However, current studies have been carried out on the matching relations between land and water resources at different levels, while there are few studies concentrating on the ternary matching relation between land, water, and grain production structure. In fact, agricultural production conditions should not be evaluated only by the matching degree of agricultural land and water resources, but also should include the adaptation of the present structure of grain production to the utilization of water resources and cultivated land resources.

In order to solve the current water crisis and ensure sustainable development of agriculture in the main grain-producing areas of North China, it is significant to explore the water–land–food nexus and offer some suggestions based on the obtained results. Therefore, this paper firstly explores the matching relationship between water and land resources for food production using the Gini coefficient and the water–land matching coefficient method. Considering the grain crop production structure, a water–land–food nexus model is constructed to evaluate the suitability and satisfaction of water, land, and grain production. Finally, the comparisons between the WL and WLF nexus is further explored for proposing possible solutions of uneven inter-regional water–land–food nexus and agricultural water and land management in the main grain-producing areas of North China.

## 2. Materials and Methods

### 2.1. Materials

#### 2.1.1. Study Area

The main grain-producing area in North China is located in the North China Plain, the second largest plain in China, with a geographical location of 32°~40° N and 114°~121° E (Figure 1). The terrain is low and flat, mostly below 50 m above sea level. It is an important grain, cotton, and oil production base in China. The study area includes the five provinces of Hebei, Shandong, Henan, Anhui, and Jiangsu, with an annual average temperature of 8~15 °C, and annual precipitation of 500~1000 mm. The crops mainly include wheat, corn, cotton, and rice. The soil is mostly brown soil and cinnamon soil, with deep soil layers and fertile soil, which is suitable for crop growth. In this study, a total of 74 prefecture-level municipal units were included, including 18 in Henan Province, 16 in Shandong Province, 16 in Anhui Province, 11 in Hebei Province, and 13 in Jiangsu Province.

It was found that the multi-year average of the total water resources in the study area from 2000 to 2020 was 23.27 billion m^3^, accounting for 7.44% of the national average total water resources, of which Hebei, Henan, Shandong, Anhui, and Jiangsu provinces were 13.98 billion m^3^, 49.18 billion m^3^, 30.42 billion m^3^, 85.99 billion m^3^, and 48.10 billion m^3^, respectively (Figure 2). The interannual average cultivated land area of the whole study area occupies 23.89% of the country. The period from 2000 to 2020 presents a cultivated land reduction, but the overall change is small, and the center of gravity of cultivated land shows a small northward shift. The total area of grain crops sown in the main grain-producing areas of the North China Plain in 2020 was 38,104.2 thousand hectares, accounting for 33% of the total area of grain cultivation in the country in that year. During the period of 2000–2020, the total planted area in the North China Plain Region was elevated by a total of 3304.6 × 10^3^ hm^2^, and the total grain output was increased by 82.062 million tons.

From 2000 to 2020, the planting area of food crops in the study area changed from 33,533 × 10^3^ hm^2^ to 38,423 × 10^3^ hm^2^, with an increase of 4890 × 10^3^ hm^2^. Among them, the planting area of rice, wheat, and maize increased by 1006, 1962, and 4945 × 10^3^ hm^2^, respectively, while the planting area of soybean and tubers decreased by 772 and 1622 × 10^3^ hm^2^, respectively. Henan and Shandong provinces mainly grow wheat and maize. Hebei province mainly grows maize and wheat, where the area of maize is larger than that of wheat. Jiangsu and Anhui provinces mainly grow wheat and rice (Figure 3). Compared with other provinces in the study area, Henan Province had the highest proportion of wheat planting area in 2020, accounting for 31.6% of the total area. Anhui Province had the highest proportion of rice and soybean planting area, accounting for 47.0% and 42.7% of the total area, respectively. Hebei Province had the highest proportion of potato planted area, accounting for 47.5% of the total area.

#### 2.1.2. Data

Grain sown area and production were obtained from the National or Provincial Statistical Yearbooks in 2000–2020. The data on cultivated land area and grain crop maturity can be obtained from the remote sensing monitoring data on the status of land use and remote sensing monitoring data on farmland maturity in China from 2000 to 2020 released by the Institute of Geographical Sciences and Resources, Chinese Academy of Sciences, respectively. Data on total water resources, surface water, groundwater, and irrigation water consumption were acquired from China Water Resources Bulletin and the Provincial or Municipal Water Resources Bulletins. The effective utilization coefficients of irrigation water are from prefecture-level municipal water resources development planning documents.

### 2.2. Methods

#### 2.2.1. The Gini Coefficient

The Gini coefficient is a vital analysis index used internationally to comprehensively examine the difference in income distribution within the population. Similar to income distribution, the spatial distribution of land and water resources is also uneven. According to Krugman’s research on the relationship between industry and space and combined with the spatial distribution characteristics of resources, the following regional Gini curves can be constructed [29]: (1) With the province as the mapping unit and the prefecture-level municipalities of each province as the basic unit, the amount of available water resources per unit of cultivated land is calculated, and the relative value is applied as the grading index of water resources matching level, and ranked; (2) The proportion of available water resources for food production and of land resource to the total water and land resources of the province in each prefecture-level city are calculated, and the cumulative proportion of water and land resources available to the total water and land resources in the province is calculated based on the ranking of (1); (3) The XY scatter plot matching the availability of available water resources for food production and the area space of cultivated land and fit the Lorentz curve is drawn. The coordinate X is the cumulative proportion of land resources in the total land resources of the province, and the coordinate Y is the cumulative proportion of available water resources for food production in the total available water resources for food production of the province; (4) The Gini coefficient is calculated. The specific calculations are as follows:(1)$$G=\frac{S_{A}}{S_{A}+S_{B}}$$
(2)$$G=1-\sum_{i=1}^{n}(X_{i}-X_{i-1})(Y_{i}+Y_{i-1})$$
where *G* is the Gini coefficient; *S_A_* is the value of the area of the region surrounded by the Lorentz curve and the absolute equality line; *S_B_* is the value of the area surrounded by the Lorentz curve and the abscissa; *X_i_* denotes the cumulative percentage of the cultivated land area in unit *i*; *Y_i_* refers to the cumulative percentage of available water resources in unit *i*; *i* is the ith calculated unit, *i* = 1, 2, *n*, and *n* is the total number of units in the study.

The regional water resources available for food production *W* is as follows:(3)$$W=\{{\begin{array}{l} W_{A}\times P\\ W_{A}-W_{B}(1-P) \end{array}}_{}\begin{array}{l} W_{A}<W_{B}\\ W_{A}\geq W_{B} \end{array}$$
where *W_A_* is the regional water resources available, 10^4^ m^3^, *W_B_* is the total regional water consumption, 10^4^ m^3^, and *P* is the proportion of irrigation water in the total regional water consumption.

The proximity of the Lorenz curve to the 45° line represents the degree of matching between the geographic distribution of cropland resources and the geographic distribution of available water resources. When the Lorentz curve completely coincides with the 45-degree line, that is, when the Gini coefficient *G* = 0, the regional distribution of land and available water resources match perfectly. Conversely, the closer the Gini coefficient *G* is to 1, the more the regional distribution of land and available water resources does not match. Table 1 shows the Gini coefficient given by the relevant UN agencies and its evaluation results.

#### 2.2.2. Water–Land Matching Coefficient Method

The agricultural water and land resource matching coefficient can represent the quantitative ratio relationship between the available water resources and the cultivated land resources that are appropriate for matching in space and time. In this study, we will use the amount of the regional water resources available for food production and the area of cultivated land available for food production to calculate the matching coefficient of water and land resources. The specific calculation is presented as follows:(4)$$R_{}=\frac{W_{}}{L}$$
where *R* is the water and land matching coefficient, 10^4^ m^3^/hm^2^, *W* is the regional water resources available for food production, 10^4^ m^3^, and *L* represents the cultivated land area available for food production, 10^4^ hm^2^.

For the area of cultivated land for grain production, it connotes the area planted with grain crops in the main grain-producing areas of the North China Plain, whereas, because of the real sowing situation, farmers will rotate their cultivated land using the crop maturity system to improve the grain crop yield while maintaining soil fertility [30]. Therefore, when determining the required cultivated land area of food crops, the current sown area of food crops needs to be divided by the maturity system of that food crop. Based on the total cultivated land area in each province and city within the North China Plain, the cultivated land area available for grain production in the region can be obtained. It can be calculated as follows:(5)$$L_{i}=\frac{L_{c}\times L_{a}}{\left(L_{c}+L_{b}\right)},L_{c}=\frac{S_{c}}{C_{c}},L_{b}=\frac{S_{b}}{C_{b}}$$
where *L_i_* is the cultivated land area available for food production in unit *i*, 10^4^ hm^2^, *L_a_* refers to the total area of regional cultivated land, 10^4^ hm^2^, *L_b_* is the actual cultivated land area used for non-food crop production, 10^4^ hm^2^, *L_c_* is the actual area of cultivated land used in food production, 10^4^ hm^2^, *S_b_* is area sown with non-food crops, 10^4^ hm^2^, *S_c_* is area sown with food crops, 10^4^ hm^2^, *C_b_* is the maturity of non-food crops, and *C_c_* represents the maturity of food crops.

#### 2.2.3. Water–Land–Food Nexus Analysis

Furthermore, this study considers that the matching degree of water and land resources possessed by regional grain production cannot completely reflect the grain production conditions. Thus, the water–land–food nexus model is developed in this study to evaluate the matching status of water resources and cultivated land resources possessed by regional grain production and the adaptation degree of current grain production structure to the utilization status of water resources and cultivated land resources. The calculation formula is written as follows:(6)$$WLF=\frac{W}{L}/\frac{\sum_{i=1}^{5}p_{i}\times M_{i}}{\left(\sum_{i=1}^{5}M_{i}/C_{i}\right)\times \eta }$$
where *WLF* indicates the water–land–food correlation coefficient, *W* refers to the amount of the regional water resources available for food production, 10^4^ m^3^, *L* is the amount of available cultivated land areas, 10^4^ hm^2^, *P_i_* is the irrigation water quota of food crop *i*, m^3^/hm^2^, *M_i_* denotes the sown area of food crop, hm^2^, η is the effective utilization coefficient of regional irrigation water, and *C_i_* is the crop system of food crop *i*.

## 3. Results

### 3.1. Equilibrium Analysis of the Matching Characteristics of Agricultural Water and Land Resources

According to the Lorenz curve of agricultural water and land resource matching and its Gini coefficient (Figure 4) for 2000 and 2020 in the five provinces of the main grain-producing areas in North China, it can be observed that the equilibrium and fairness of water and land matching in Jiangsu province are significantly better than those in other provinces in the study area. Therefore, by judging the water and land matching of the five provinces in the study area from the perspective of the equilibrium of water and land matching of municipalities in the province, it can be found that the distribution of available water resources for grain production and cultivated land resources in Jiangsu province has a high equilibrium, and the distribution of water and land resources in the province is reasonable. Approximately 50% of available water resources for grain production in Hebei province supplies 70% of cultivated land in 2020. The unbalanced distribution of water and land resources in Henan Province has deteriorated, with nearly 30% of the available water resources for food production supplying 65% of the cultivated land by 2020. Additionally, the available water resources for food production are mostly concentrated in Xinyang, Zhumadian, and Nanyang. There was an increase in the unbalanced distribution of water and land resources in Anhui Province from 2000 to 2020. By 2020, approximately 30% of the available water resources for food production supplies 70% of the cultivated land, while the remaining 70% of the available water resources for food production only supplies less than 30% of the cultivated land, and the available water resources for food production is concentrated mostly in Huangshan, Lu’an, Xuancheng, and Chizhou. Thus, the distribution of water and land is extremely unbalanced, causing the Gini coefficient to cross the warning line of 0.4. In Shandong province, the Gini coefficient of matching water and land resources decreases during the study period and returns to a relatively reasonable range. This is mainly due to the increase in the amount of water available for food production in cities including Liaocheng, Dezhou, and Weifang. Therefore, the establishment of new water conservancy projects and the development of water conservation techniques through scientific and technological development are more favorable for improving the balance of water and land resources.

The mean values of Gini coefficients of land and water resources in Henan, Hebei, Anhui, Shandong, and Jiangsu Province in the study area from 2000 to 2020 were 0.38, 0.34, 0.53, 0.29, and 0.20, respectively, and Hebei and Jiangsu Province were within the reasonable range of land and water resource allocation, whereas Anhui Province had a large unevenness of land and water resource allocation and was beyond the reasonable range (Figure 5). The Gini coefficient of land and water matching can only characterize the fairness and reasonableness of water and land resource allocation within the province. Therefore, from this perspective, if there exists large unevenness of water and land resource allocation within the province, priority should be given to intra-provincial inter-city water transfer from cities with more water resources available for food production to other cities suffering from water scarcity in the province. For those cities and provinces with water and land matching Gini coefficients within a reasonable range but that are still short of water in the provinces with more reasonable water and land distribution, consideration can be given to transferring water from outside the province or optimizing the food crop planting structure to resolve the problem of mismatching water and land resources.

### 3.2. Changes of Water–Land Nexus Available for Food Production

According to the matching coefficient of water and land resources calculated based on the amount of water resources available for food production and the area of cultivated land available, from the perspective of prefecture-level cities—concerning spatial and temporal distribution, from 2000 to 2020—the high-value areas of the matching coefficient of water and land resources in the main grain-producing areas in North China are mostly in Huangshan, Chizhou, Xuancheng, Anqing, Lu’an, Yancheng, Huai’an, Xinyang, Nanjing, and Lianyungang. The second highest value areas are in Nanyang, Taizhou, Suqian, Changzhou, Xuzhou, Wuhu, Zhumadian, and Wuxi. The high-value and secondary high-value areas are mainly in Jiangsu Province, southern Anhui Province, and northern Henan Province. Additionally, the low-value areas of land and water resource matching coefficient are mainly in Hengshui, Cangzhou, Langfang, Zhangjiakou, Zhengzhou, and Dongying. The low-value areas are mainly distributed in southern Hebei Province and central Shandong Province. The spatial and temporal changes show that the interannual changes from 2000 to 2020 are more obvious. Compared with 2000, 42 cities in the main grain-producing areas of North China present a decrease in the matching coefficient of land and water resources in 2020, among which Huai’an, Yancheng, Lianyungang, Taizhou, Nanjing, Yangzhou, and Nanyang exhibit a greater decrease, mainly in central Henan Province and eastern Jiangsu Province. Totally 32 cities have a decrease in the matching coefficient of land and water resources, including Huangshan, Chizhou, Anqing, Tongling, and Xuancheng, which are mainly located in the south-central part of Anhui Province.

Generally, the spatial and temporal distribution of water and land resource matching in the main grain-producing areas of the North China Plain is uneven, with significant differences, revealing a pattern of “the best in the southwest, the second best in the central part, and the worst in the north”. From 2000 to 2020, the overall trend of water and land resource matching coefficient in the north is decreasing, and the degree of water and land resource matching in the north is much worse than that in the south (Figure 6). Combined with the cultivated land-grain management relationship and the amount of water available for grain production under the natural background, the areas with better matching water and land resources in the main grain-producing regions of North China are mainly those with abundant water resources available for grain production. However, the scarcity of water resources available for grain production in Hebei Province and northwestern Shandong Province is the main reason for their low matching coefficients. Therefore, external water transfer or the development of new water-saving technologies can be considered to increase the amount of water available for grain production. Meanwhile, the matching coefficient of water and land resources is also constrained by the available cultivated land area. With the largest cultivated land area, Henan Province has taken up more production functions in grain sowing, and the grain sowing area has increased by 51.72% from 2000 to 2020. Nevertheless, the amount of water available for grain production has not increased simultaneously with the dramatically rapid growth in demand for grain production, thus presenting a reducing matching degree of water and land. Therefore, it is significant to improve the state of low water–land matching by optimizing the cultivation structure of food crops.

### 3.3. Spatial and Temporal Change Characteristics of Water–Land–Food Nexus

By adopting the water–land–food nexus model, this study investigates the matching degree of available water and cultivated land resources for food production and the adaptation of food production structure to the actual utilization of water and cultivated land resources. Furthermore, it evaluates the suitability and satisfaction degree of water and land resources for food production. When WLF < 0.55, the degree of matching is extremely poor, indicating that the matching status of available water resources and cultivated land resources for food production in the region cannot satisfy the agricultural water demand of food production under the current scale of cultivated land use, and is in a state of water scarcity. When the WLF is between 0.55 and 0.85, the match is poor, implying that the amount of water available for food production in the region can barely support the agricultural water demand of the food production structure under the current scale of cultivated land use, while the water resources are still in shortage in relative terms. When WLF is between 1.25 and 1.55, the degree of matching is good, which suggests that the matching between the amount of water available for regional food production and cultivated land resources can better meet the agricultural water demand of the food production structure under the current scale of cultivated land use. When WLF ≥ 1.55, the matching degree is the best, which means that there are still abundant water resources and sufficient water resources while satisfying the agricultural water demand of food production structure under the current scale of cultivated land use.

Based on the spatial and temporal changes of the water–land–food nexus in the study area from 2000 to 2020 (Figure 7), it can be found that the cities with water–land–food matching degrees above the standard level decreased from 43.2% in 2000 to 39.2% in 2020, and only 20.3% of the cities with WLF above 1.25 in 2015. During the study period, the cities with a consistently better water–land–food matching degree included Nanyang, Zhumadian, and Xinyang in Henan Province, Rizhao and Zaozhuang in Shandong Province, and Huangshan, Chizhou, Xuancheng, Anqing, Lu’an, Fuyang, and Bozhou in Anhui Province, while most cities in Hebei Province and northwestern Shandong Province were in a state of water scarcity, and large areas of cultivated land with the irrigation needs of food crops could not be satisfied. Hebei Province and northwestern Shandong Province are constrained by the scarcity of available water resources for food.

From 2000 to 2020, the matching degree between water, land, and food in the study area is not only influenced by the amount of water available for grain production, but also by the difference in the structure of food crops between regions. In addition to the decrease in water resources available for grain production, the proportion of wheat cultivation in western Jiangsu Province and the proportion of maize cultivation in central Henan Province increase significantly, resulting in an increase in water demand, making the water resources available for grain production unable to meet their grain production needs. Hebei Province and northwestern Shandong Province are constrained by the scarcity of water resources available for food on the one hand, and the poor matching of water, land, and food due to the cultivation of wheat and corn, which are high-water-demand crops, on the other hand. In central Anhui Province, the water–land–food matching degree in 2020 is clearly on a positive trend, mainly influenced by the increase in precipitation in that year. Yantai and Weihai in Shandong Province have a better water–land–food matching degree, mainly due to the decrease in the proportion of wheat, a water-consuming crop.

Generally, the water–land–food matching degree in the main grain-producing areas of North China tends to decrease, and 60.8% of the cities are imbalanced or mildly imbalanced until 2020. The overall situation of the water–land–food matching degree is characterized as “worse in the north and better in the south”; most cities in Hebei Province and the northwestern part of Shandong Province are characterized by water shortage, and the natural background of regional water resources alone cannot effectively support their food production. The results suggest that the water–land–food matching degree in the main grain-producing areas of North China can be further improved by enhancing water utilization efficiency and optimizing grain cultivation structure.

### 3.4. Relationship between Water–Land Nexus and Water–Land–Food Nexus

In this paper, we further characterize the WL and WLF matching degrees of 74 prefecture-level cities in the study area with scatter diagrams. We divided the comparison results into four categories, with the top left, bottom left, top right, and bottom right quadrants, respectively, representing high WL-low WLF, low WL-low WLF, high WL-high WLF, and low WL-high WLF (Figure 8).

The comparison revealed that the number of prefectures with low WL-low WLF increased during the study period, with 59 prefectures having poor water–land matching degree and water–land–food matching degree in 2010, and the number of prefectures with high WL-high WLF decreased, with only 7 prefectures having good water–land matching degree and water–land–food matching degree in 2020. Among them, Lu’an, Anqing, Xuancheng, Chizhou, and Huangshan were in the high WL-high WLF quadrant during the study period, indicating that their water–land matching degree and water–land–food matching degree have been better; Dongying, Baoding, Tangshan, Anyang, Langfang, Zhangjiakou, Dezhou, Xinxiang, Cangzhou, Jinan, Zibo, Binzhou, Weifang, Puyang, Yantai, Jiaozuo, Shijiazhuang, Qinhuangdao, Liaocheng, Hengshui, Xingtai, Handan, Zhengzhou, Qingdao, and Hebi were in the low WL-low WLF quadrant during the study period, indicating that the WL and WLF in these 26 prefecture-level cities were poor.

From 2000 to 2020, Hefei, Huainan, Wuhu, and Tongling changed from low WL-low WLF to high WL-high WLF zones due to the increase in local precipitation in 2020, which led to an increase in available water resources for grain production. The change from high WL-high WLF to low WL-low WLF in Nanjing, Pingdingshan, Yangzhou, and Luohe is related to the increase in the proportion of wheat, which is a high-water-consuming crop; this increase in crop water demand results in larger comprehensive irrigation quotas for food crops and a decrease in the amount of water available for food production. By 2020, 14 cities have a better WLF despite the water and land resource mismatches, such as Zhumadian, Zhoukou, and Shangqiu in Henan Province, and Fuyang, Bozhou, and Maanshan in Anhui Province, indicating that these cities have a high water use efficiency, probably because they have further developed modernized water-saving and intensive agricultural production methods, and the amount of water available for grain production can meet the water requirements of crops. In addition, Suqian, Changzhou, Wuxi and Huai’an cities have a higher WL but a lower WLF. Suqian, Changzhou, Wuxi and Huai’an have a high WL and a low WLF, which indicates that these cities have a relatively crude utilization of water resources and do not pay attention to the technology development and capital investment of agricultural water-saving potential, thus, the water resources utilization efficiency is low, and the water use efficiency of grain irrigation is not high. Coupled with the larger cultivation area of wheat and rice, this results in the irrigation water demand also being larger, leading to the low WLF. Therefore, it is possible to improve agricultural water use efficiency and reduce irrigation water demand to a certain extent by appropriately increasing the corn sowing area or changing the maturity of food crops while ensuring the normal production of other food crops such as rice and wheat. At the same time, we can also further increase the financial investment and technology research and development for water-saving agricultural technology.

## 4. Discussion

The overall upward trend of the Gini coefficient during the study period in the NCP indicates an increasing imbalance of the water–land matching degree, which does not contradict the findings of Li et al. who found that the coupling degree of water–energy–land–food (WELF) in most provinces in China is at a barely coordinated stage and in a decreasing trend [31]. The results of the Gini coefficient demonstrate that the distribution of water and land resources in Anhui Province is poorly balanced, while its water and land match is good, revealing the spatial imbalance of water and land within Anhui Province. Even though the water resources in Anhui are in good condition compared with the north of the study area, the spatial distribution of the internal water and land resources is very uneven. Additionally, the spatial distribution of the internal water and land resources between the regions varies significantly, for instance, Huangshan has a good water–land matching degree because it has abundant water to supply very little cultivated land, while Maanshan has a poor water–land matching degree due to having less cultivated land and very little water. Moreover, the results of the water–land and water–land–food matching degree in this study are basically the same, and the areas with high values of water–land matching degree and water–land–food nexus are mostly concentrated in the central and southwestern parts of the main grain-producing areas in North China. The high-value area is in the northern part of the study area, while the southern part is the low-value area. Thus, the south and central parts of the study area have a better matching degree of water–land and water–land–food compared with the whole study area. Nevertheless, there are also regions where the results are different, for example, Nanyang, Sanmenxia, Weihai, and Zhumadian, where the water–land matching degree is average, but the water–land–food nexus is good; this is because the combined irrigation quotas for food crops are small and crops with low water requirements are grown in these places. The result is supported by Yang et al.’s conclusion that some cities in Henan Province, such as Zhumadian and Nanyang, have low crop water requirements [6]. The water available for grain production in these places is better capable of meeting the water requirements for the same grain sowing under the same cultivated land conditions. There are studies showing that Henan, Shandong, and Hebei province in the study area are water-scarce provinces [18], which provides some strong supporting evidence for the reliability of the conclusions of our results: in general, from 2000 to 2020, the water–land–food matching degree in the region is influenced by the abundance of water resources available for food production, and the overall matching degree is higher in the south than in the north.

Therefore, referring to the results of the unbalanced distribution of water and land resources evaluated by the Gini coefficient, the following recommendations are proposed as: (1) areas with more natural precipitation can strengthen the construction of water conservancy, reasonably saving the available surface water resources. (2) Strengthen the connection of water systems of rivers, lakes, and reservoirs between regions, as well as enhance the water resources storage capacity of each region; according to the unbalanced distribution of water and land resources of most cities in Anhui Province; priority can be given to intra-provincial water transfer by transferring water from the nearest city. (3) In areas with the unbalanced distribution of water and land, water resources can be transferred between each other, for instance, engineering or non-engineering measures can be used to divert water resources from abundant areas to neighboring water shortage areas in order to promote the balanced distribution of water resources in each area. Based on the results of the matching degree of water–land and water–land–food, the following suggestions are made: (1) Low WL-low WLF and high WL-low WLF areas should further explore the optimization of grain cultivation structure, develop low water-consuming and high-yielding crop varieties, and reduce the irrigation water intensity. The wheat cultivation area in low-yielding and low-efficiency areas can be suppressed. Meanwhile, on the basis of stabilizing vegetable prices and improving vegetable quality, the scale of water-consuming vegetable cultivation can be moderately reduced, and the proportion of maize cultivation should be appropriately increased. At the same time, the wheat–maize biannual system can be changed to mono-annual or a two-year three-cropping system that is suitable for water cultivation or develop semi-dry land agriculture in suitable areas [32,33]. In addition, they can support field water-saving irrigation facilities for grain fields in water-scarce areas or reasonably develop and utilize transit surface water resources to improve water utilization efficiency [34,35,36]. It is also essential to adhere to the government’s water conservation and water use target control, combine with market mechanisms, reasonably formulate agricultural water pricing policies and other sustainable improvements in agricultural water use efficiency, gradually eliminate the unfavorable factors that restrict unbalanced development between regions, as well as promote balanced regional development [37,38,39]. (2) Low WL-high WLF areas can maintain or further improve the current water use efficiency and try to expand irrigated areas with full consideration of the carrying capacity of water and land resources and optimal allocation of water resources. In the process of implementing territorial spatial planning in the new era, all departments should also pay attention to further improving the level of coordination between cultivated land and water resources, aiming to achieve optimal allocation [40,41]. (3) For high WL-high WLF areas, we should rely on their advantageous conditions to further strengthen the protection of the “quantity, quality, and ecology” of cultivated land, carry out the Green Transition of Cultivated Land-use in the region, and pay attention to use fertilizers and pesticides scientifically to reduce environmental pollution—especially soil pollution [42,43]. In addition, we should further strengthen the protection of farmland ecosystems, enhance the biodiversity of cultivated land and the stability of farmland ecosystem, and highlight their important role in ensuring national food security.

The main grain-producing areas in North China are vital grain production bases in China. Previous studies have mostly employed the amount of agricultural water resources per unit area of cultivated land to characterize water–land matching degree, thus ignoring the influence of regional grain crop cultivation structure and other factors. In this study, to locate the water–land matching degree of grain production more precisely, we adopt cultivated land area and water resources available for food production, which can thus eliminate the influence of inter-regional water distribution and transit water use. Additionally, this study breaks through the original binary perspective on water and land resources and introduces the water–land–food nexus model to further consider the adaptation of the current grain production structure to the utilization of water resources and cultivated land resources, as well as evaluate the suitability and satisfaction of water resources, land resources, and food production. Moreover, this study provides a theoretical basis for promoting the optimal allocation of water–land–food in the main grain producing areas of North China.

The integrated study of water, land, and food is a complex scientific task, and it covers the separation and combination of time and space at different research scales, including regional, provincial, and municipal scales. Although this study has made some progress, there remain some shortcomings. The water resources discussed in this study are the water resources available for food production in the region based on the natural background conditions. Further, the constraints of the water use red line on the amount available for food production have not been considered. The indicators of total water use control should be further considered. In addition, the matching status and development potential of agricultural water resources guided by water use planning need to be explored. This study concentrates on the matching degree of water resources, cultivated land resources, and food production in a certain region, and can be further refined to the microscopic evaluation of the suitability of land use in different areas as well as the adaptability of water demand of different crops, hoping to optimize the spatial and temporal allocation of water and land resources at small and medium scales in watersheds or counties, which will be more practical. The analysis of water–land and water–land–food matching degree in this paper are only based on the existing spatial and temporal data, and the subsequent systematic study of the allocation efficiency of water–land and water–land–food and the analysis of the specific factors influencing their allocation efficiency will provide the foundation for the improvement of their allocation efficiency, which will be the following direction to be strengthened.

## 5. Conclusions

In this paper, the Gini coefficient and water–land matching coefficient methods are adopted for investigating the water–land matching degree in the main grain-producing areas of North China from 2000 to 2020, and the water–land–food nexus model is further constructed to explore the water–land–food matching degree in the five provinces of the main grain-producing areas of North China and its 74 municipalities from 2000 to 2020. The main conclusions are presented as follows:

1. In general, the Gini coefficient presents an increasing trend in the main grain-producing areas in North China from 2000 to 2020, indicating an increasing imbalance in the matching degree of agricultural water and land resources among inter-regions. Jiangsu Province has the most balanced distribution, whereas the imbalance between regions in Anhui Province is significant, with a multi-year average Gini coefficient of 0.53.

2. The regional coordination level of water and land resources in the main grain-producing areas of North China varies significantly during 2000–2020, with the northern area obviously inferior to the southern area year by year, revealing a pattern of “the best in the southwest, the second best in the central part, and the worst in the north”, with the northern part being more or less 0.2 million m^3^/hm^2^, the central part being mostly in the range of 0.2–0.85 million m^3^/hm^2^, and the southern region being greater than 0.85 million m^3^/hm^2^. In particular, most cities in Hebei Province and the northwestern part of Shandong Province are characterized by water shortage, and the natural background of regional water resources alone cannot effectively support their food production.

3. Using the water–land–food nexus analysis, the matching degree of more than 60% of the municipalities in the main grain-producing areas in North China from 2000 to 2020 are uneven. The overall situation of water–land–food matching degree is characterized by “worse in the north and better in the south”, and generally, the imbalance in the water–land–food nexus between regions and provinces in the main grain-producing areas of North China is extremely significant.

4. There are significant gaps in the WL and WLF nexus in different regions. The cities belonging to the low WL-low WLF and high WL-low WLF quadrant should be considered as key targets when formulating policies. For these areas, it is vital to optimize the grain cultivation structure adapting to water resources distribution, promote semi-dryland farming, develop low water-consuming and high-yielding crop varieties, and reduce the irrigation water intensity. Additionally, especially the northern part of the study area can appropriately change the wheat–maize biannual system to mono-annual or a two-year three-cropping system, which is suitable for water cultivation, and reasonably develop and utilize transit surface water resources. Meanwhile, it is also essential to adhere to the government’s water conservation and water use target control to improve agricultural water use efficiency.

## Figures and Tables

**Figure 1 foods-12-00712-f001:**
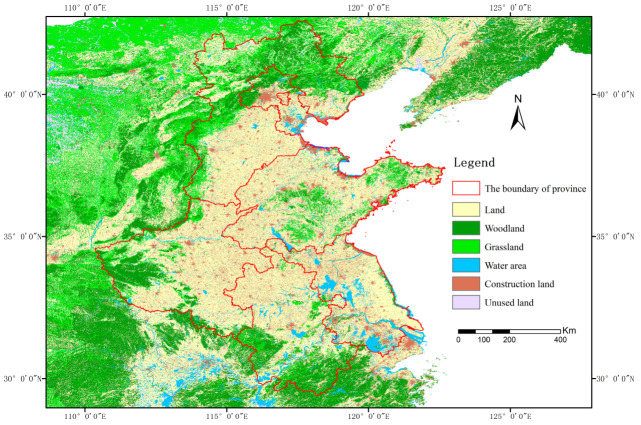
The location of study area and its land use.

**Figure 2 foods-12-00712-f002:**
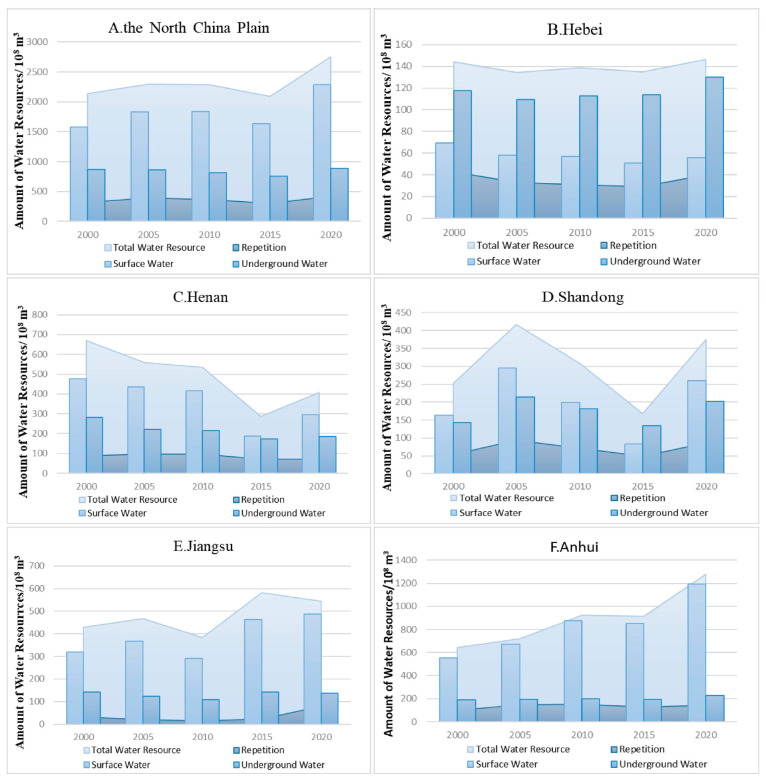
The changes in total water resources and their composition from 2000 to 2020 in the main grain-producing area of North China (**A**), Hebei (**B**), Henan (**C**), Shandong (**D**), Jiangsu (**E**), and Anhui (**F**) province.

**Figure 3 foods-12-00712-f003:**
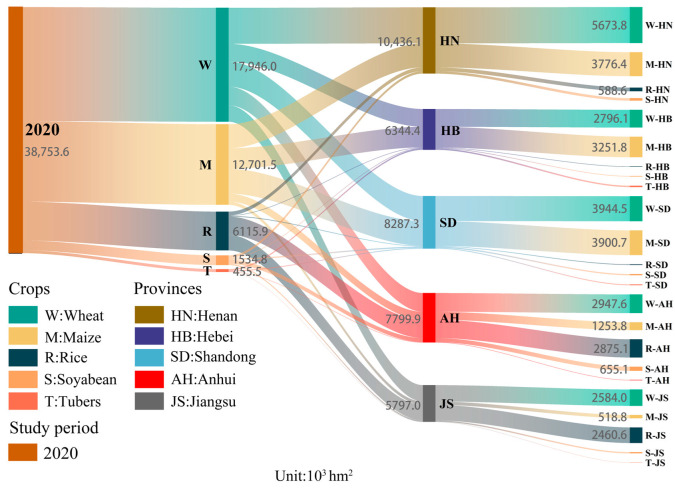
The structure of food crops in 2020 in the main grain-producing area of North China.

**Figure 4 foods-12-00712-f004:**
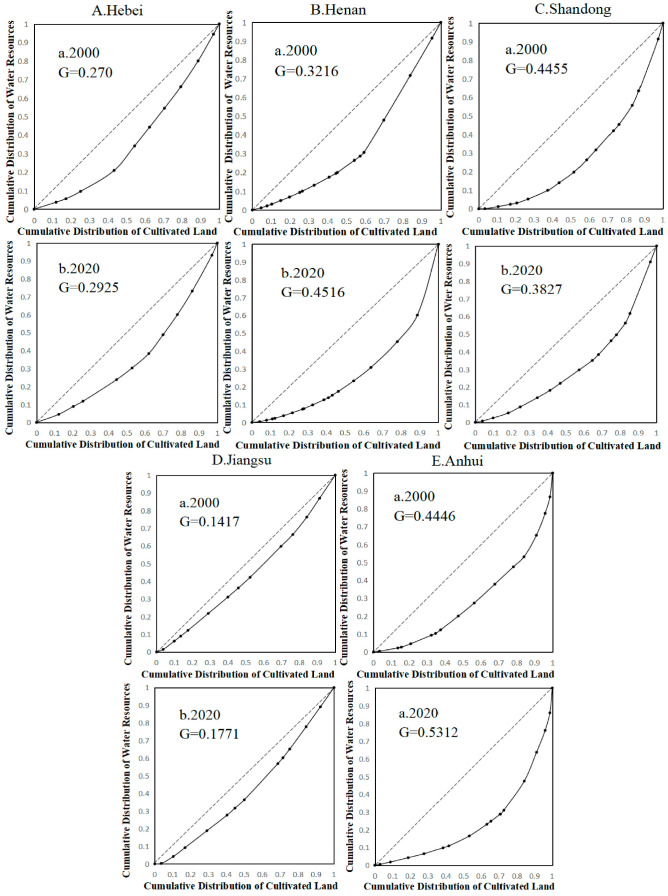
Lorentz curves of matching the amount of water available for food production with the cultivated land resources in Hebei (**A**), Henan (**B**), Shandong (**C**), Jiangsu (**D**), and Anhui (**E**) provinces. The dashed line is absolute average line.

**Figure 5 foods-12-00712-f005:**
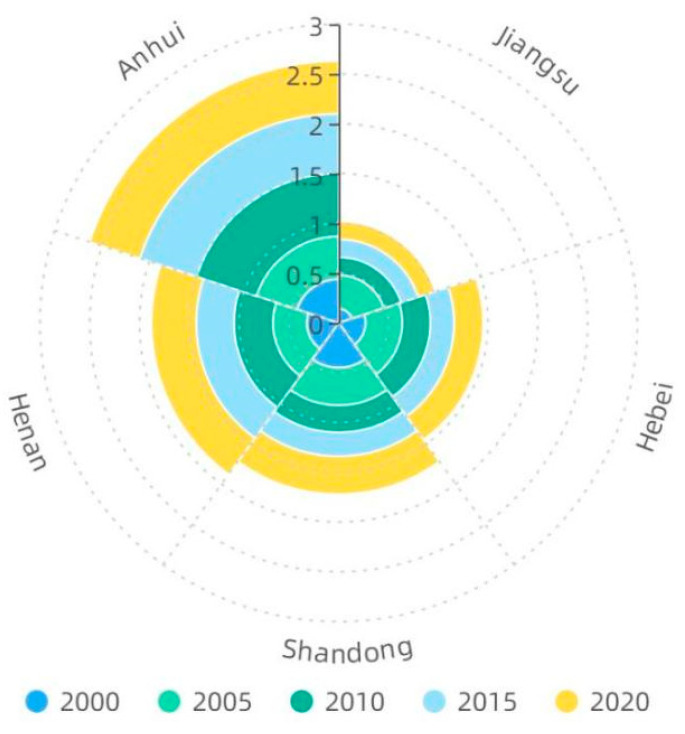
The change of the Gini coefficient between cultivated land resources and the water resources for agriculture.

**Figure 6 foods-12-00712-f006:**
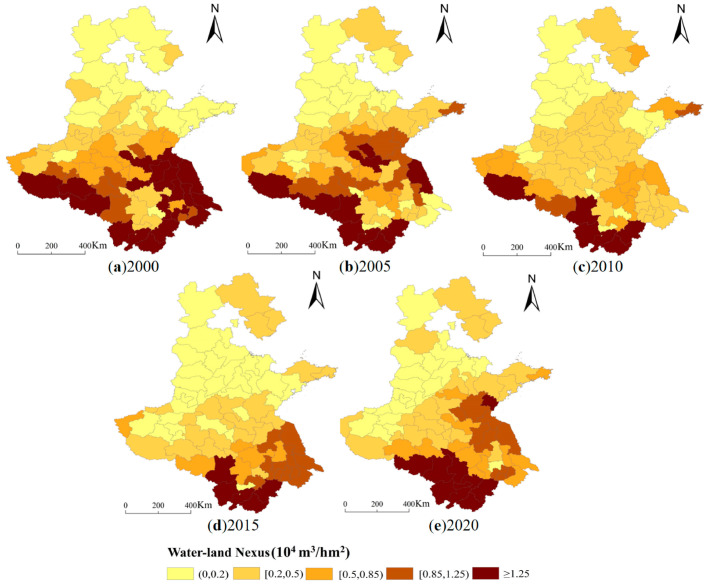
Spatial and temporal variation of the water–land nexus. (**a**) 2000 (**b**) 2005 (**c**) 2010 (**d**) 2015 (**e**) 2020.

**Figure 7 foods-12-00712-f007:**
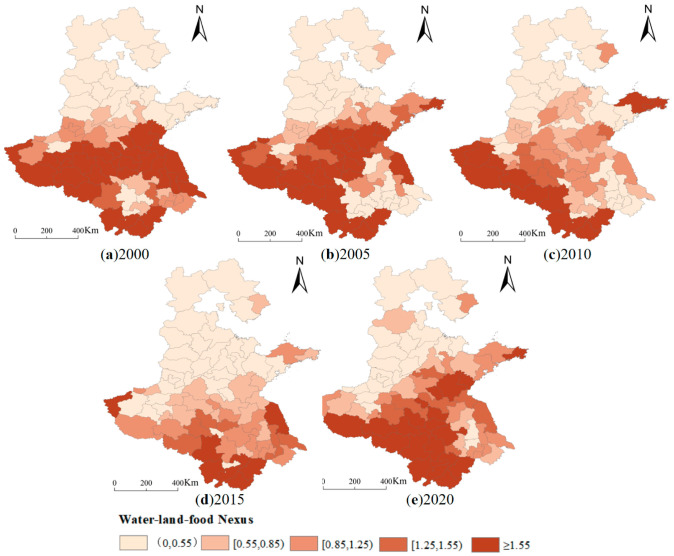
Spatial and temporal variation of the water–land–food nexus. (**a**) 2000 (**b**) 2005 (**c**) 2010 (**d**) 2015 (**e**) 2020.

**Figure 8 foods-12-00712-f008:**
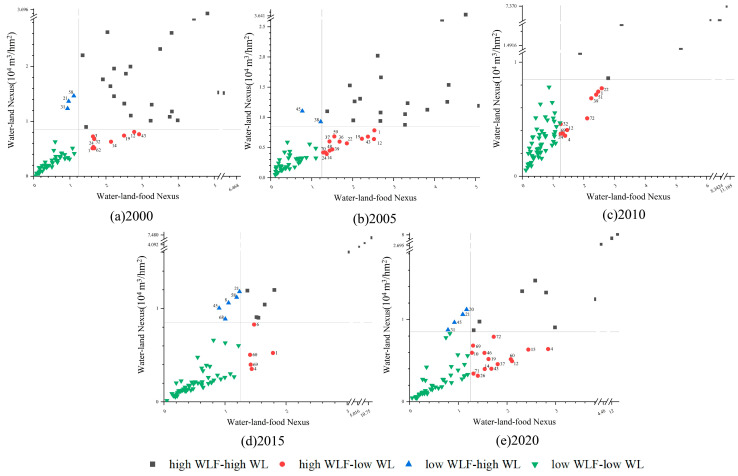
Water–land nexus and water–land–food nexus of 74 cities in the main grain-producing areas of North China. (**a**) 2000 (**b**) 2005 (**c**) 2010 (**d**) 2015 (**e**) 2020. All input cities and theircode numbers are shown in Appendix A.

**Table 1 foods-12-00712-t001:** Gini coefficient and evaluation results.

Gini Coefficient	<0.2	0.2~0.3	0.3~0.4	0.4~0.5	>0.5
Evaluation results	Absolute average	Comparative average	Relatively reasonable	Large gaps	Disparity

## Data Availability

The dataset used and analyzed during the current study are 670 available from the corresponding author on reasonable request.

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
