# Peer review of "Water–Land–Food Nexus for Sustainable Agricultural Development in Main Grain-Producing Areas of North China Plain"

_foods, 2023, doi:10.3390/foods12040712_

Round 1
Reviewer 1 Report
That is a regional study that makes use of a dedicated collection of territorial data and relevant information on the availability of resources for agriculture. It proceeds with very interesting analyses and characterizes a study with high applicability power for territorial actions and policies aimed at sustainability in the face of a challenging context of demands for food production and scarcity of land and water resources. Overall, it can be considered as a very original contribution, especially for the applicability of the nexus thinking perspective to a specific territorial context, in this case, the northern region of China. The applied model can be considered for its replicability and the conclusions are quite pertinent, mainly indicating relevant aspects of territorial imbalances and the need for appropriate alternatives for sustainable development and for coping with iniquities in terms of scarce resources.
Although the manuscript does not propose to dialogue in a broader sense with the ecosystem services that support agricultural production in the provinces studied, I think there should be some comment only as a caveat to the way it considered the cultivated land available. In fact, it would be important to point out a bit about the conservation of the ecosystems in the territories studied in view of the fact that it would not be possible only to count on the conversion of the territory into land for cultivation.
In the discussions, I found a lack of dialogue and comparative reflection with other studies that have done similar analyses involving nexus thinking on regional contexts and the availability of water and land for food production. This kind of dialogue with literature could make the discussions and conclusions more robust.
Besides these comments, I suggest a review of some typing errors, like missing spaces between words and punctuation errors. For example: line 56 “by 2050(FAO, 2021)”; line 82 “However, groundwater there has been” it seems that ´there´ is not necessary here; line 94 “Liu et al., 2018).Based”; line 333 “resolvethe”; line 554 “can be suppressed,.Meanwhile,”.
Reviewer 2 Report
The research presented in the manuscript is very good and relevant in the field and I congratulate the authors for it. However, the presentation is one that does not support, in this form. It is extremely difficult to read the material because its structure is not coherent, it is almost impossible to identify the elements of authenticity versus the elements of documentation.
Although the paper must be rewritten and the sections and their content reformulated, I will recommend it only after reformulating the manuscript and reconsidering after major revision.
Below are my comments, point by point, which I have notified and in the PDF format of the manuscript, which I am attaching.
1. The manuscript does not use the FOODS Journal template.
2. The abstract should be a total of about 200 words maximum.
3. The abstract should be an objective representation of the article and should not exaggerate the main conclusions.
4. The expression "security of national food" is correct. It refers to the established term "national food security".
5. In the text, reference numbers should be placed in square brackets [ ], and placed before the punctuation; for example [1], [1–3] or [1,3].
6. This paragraph is useless: "Consequently, the paper is structured into 6 sections. Section 1 is the introduction. Section 2 introduces the study area and data. Section 3 presents the research methods. Section 4 presents results and analyses about matching characteristics of water and land resources represented by the Gini coefficient, matching relations between agricultural water and land resources, the water-land-food nexus, and the comparisons between WL and WLF. Section 5 discusses the study results, and Section 6 draws the conclusions."
7. The Research Manuscript Sections do not meet the journal requirements.
8. All the titles of the figures does not meet the journal requirements.
9. The details must be presented, if applicable: "Informed Consent Statement: Informed consent was obtained from all subjects involved in the study."
10. References - It must be structured, according to the requirements of the journal, and the date of access must be mentioned, where appropriate.

Round 2
Reviewer 2 Report
All my observations were accepted by the authors, and the mentioned deficiencies were rectified.